

# Variability of Bulk Water Vapor Content in the Marine Cloudy Boundary Layers from Microwave and Near-Infrared Imagery

Luis F. Millán[1], Matthew D. Lebsock[1], and Joao Teixeira[1]

[1]Jet Propulsion Laboratory, California Institute of Technology, Pasadena, California, USA

*Correspondence to:* L. Millán (luis.f.millan@jpl.nasa.gov)

**Abstract.** This work uses the synergy of collocated microwave radiometry and near-infrared imagery to study the marine boundary layer water vapor. The Advanced Microwave Scanning Radiometer (AMSR) provides the total column water vapor, while the Moderate Resolution Imaging Spectroradiometer (MODIS) near-infrared imagery provides the water vapor above the cloud layers. The difference between the two gives the vapor between the surface and the cloud top, which may be interpreted as

the boundary layer water vapor under certain conditions. As a by product of this algorithm, we also store cloud top information of the MODIS pixels used, a proxy for the inversion height, as well as the sea surface temperature and total column water vapor from the AMSR measurements. Hence, the AMSR-MODIS dataset provides several of the variables associated with the boundary layer thermodynamic structure. Comparisons against radiosondes, and GPS-Radio Occultation data demonstrate the robustness of these boundary layer water vapor estimates. We explore the annual cycle of the number of observations as a

proxy for stratus cloud amount, in well known stratus regions; we then exploit the 16 years of AMSR-MODIS synergy to study for the first time the annual variations of the boundary layer water vapor in comparison to the sea surface temperature and the boundary layer cloud top height (equivalent to the inversion height) climatologies, and lastly, we explore the climatological behavior of these variables on stratocumulus-to-cumulus transitions.

## 1    Introduction

The boundary layer may be defined as the lower part of the troposphere that is directly influenced by the presence of the Earth's surface through turbulence. This layer mediates the exchanges of energy, momentum, water, carbon, and pollutants between the surface and the rest of the atmosphere and responds to surface forcing with a timescale of about an hour or less (Stull, 1988). Further, boundary layer processes are also intimately coupled with low clouds, such as stratocumulus. Stratocumulus

are the most common cloud type covering around one-fifth of the Earth's surface (with mostly four-fifths of them located over the ocean) and thus have a profound impact on Earth's energy balance, primarily through solar radiation reflection (e.g., Wood, 2012). As such, boundary layer processes are crucial for understanding cloud-climate feedback mechanisms (e.g., Teixeira et al., 2011).

Despite their importance, boundary layer process are still not well represented in weather and climate models. For example,

differences in the response of low clouds to warming scenarios are responsible for most of the spread in model-based estimates





of equilibrium climate sensitivity (Bony and Dufresne, 2005; Randall et al., 2007) and this spread appears to be attributable to how cloud, convective, and boundary layer processes are parameterized in such models (Boucher et al., 2013). However, one major issue in the development of accurate boundary layer parameterizations is the lack of global measurements.

The aim of this study is to show results from a ~16 year boundary layer column water vapor (BL-CWV) dataset derived
from the synergy of microwave and near-infrared imagery. Near-infrared imagery provides the water vapor above the clouds (by measuring the solar radiation reflected near the 0.94-$\mu$m water vapor band) while microwave radiometry provides information on the total column water vapor (by measuring at the water vapor absorption line near 22 GHz). As shown by Millán et al. (2016), the difference between their water vapor information provides an estimate of the BL-CWV when the cloud top is capped at the boundary layer top.

Variability in the boundary layer water vapor plays an important role in the evolution of clouds and precipitation. Some field campaigns (e.g., Crum and Stull, 1987; Weckwerth et al., 1996, 2004) have provided some information about its temporal and spatial distribution in a few regions but its global variability and impact on clouds is still not properly understood. For example, subtle fluctuations in the vertical profile of water vapor appear to be associated with recurring stratocumulus and cumulus regimes (Betts and Boers, 1990). Further, several studies have shown that boundary layer water vapor is a critical quantity
required for forecasting the initiation of convection (Crook, 1996; Ziegler and Rasmussen, 1998; Fabry, 2006; Martin and Xue, 2006). The combination of microwave and near-infrared imagery provides a unique capability to study the column water vapor in the planetary boundary layer.

## 2   Measurements

In this study, the AMSR-MODIS BL-CWV dataset version 2 is used. This dataset was produced merging passive microwave
and near-infrared CWV measurements as part of a NASA Making Earth System Data Records for Use in Research Environments (MEaSURES) project. In short, BL-CWV was found by subtracting the CWV above the clouds estimated by the Moderate Resolution Imaging Spectroradiometer (MODIS) from the total CWV estimated by Advanced Microwave Scanning Radiometer (AMSR) instruments. In particular, we use AMSR-E, AMSR-2 and AQUA MODIS data which allow us to estimate the BL-CWV from 2002 to date; except for a gap between April 2011 and July 2012 when AMSR-E stopped operating
and AMSR-2 became operational.

The AMSR instruments are dual-polarized conically scanning microwave radiometers with channels measuring in between 6.9-89 GHz. They provide day and night estimates of total CWV over the oceans with an estimated error of ~0.6 mm (Wentz and Meissner, 2000). Through-out this study we used the Remote Sensing Systems (REMSS) CWV retrievals, in particular version 7, which aggregates these estimates to a quarter-degree spatial resolution. MODIS is an imaging spectroradiometer
with 36 channels spread through-out the visible, near-infrared, and infrared. Here, we use version 6.0 except during December when cloud top height values were found to be unphysically large and inconsistent with the other months [R. Frey, Personal Communication]. Instead, version 6.1 was used for all December months. In particular, we use the CWV estimated using near-infrared channels that have an estimated error between 5% and 10% (Gao and Kaufman, 2003).





All these instruments orbit in tandem measuring the same volume of air within minutes of each other, that is, by design, these measurements are collocated. The MODIS retrievals of above cloud water vapor have poor height registration when the cloud is either thin or broken. To alleviate these biases several flags as well as proximity tests are applied to remove pixels with intrapixel heterogeneity and/or high clouds as specified by Millán et al. (2016). That is, we aim to identify homogeneous fields

of liquid clouds in the MODIS data. Version 2 is the second public release of the AMSR-MODIS data. The only difference against version 1 is that high clouds are masked out using the cloud phase optical properties. We only use the clouds which phase has been identified as liquid by the cloud thermodynamic phase classification algorithm (Platnick et al., 2015). Version 1 instead screened only pixels where cirrus or aerosols were detected using the 1.38-$\mu$m high-cloud flag (MYD35).

During the processing, the algorithm uses the MODIS level 2 products in their native grid (i.e. MODIS pixels with a 1 km

size at nadir) before binning the data into a 1° by 1° grid. We produce daily and monthly files. Figure 1 shows an example of a BL-CWV daily as well as a monthly composite. It also shows its associated standard deviation as well as the number of the number of single observations (MODIS pixels) used in each grid. Note that, as a by product of the BL-CWV algorithm, we also save the cloud top height (BL-CTH), the cloud top pressure (BL-CTP) and the cloud top temperature (BL-CTT) of the MODIS pixels used, as well as the sea surface temperature (SST) and total CWV from AMSR in the same grid. As such, the

AMSR-MODIS dataset provides several of the variables associated with the bulk boundary layer thermodynamic properties. Monthly files were constructed aggregating the daily files neglecting pixels which daily standard deviation was greater than 0.2 cm. This threshold mostly rejects pixels in the intertropical convergence zone (ITCZ) where the boundary layer is not well defined.

## 3  Comparisons with other observations

In this section the accuracy of the AMSR-MODIS V2 BL-CWV measurements is assessed through comparisons with radiosondes and Global Positioning System Radio Occultation (GPS-RO) measurements. For these comparisons, we consider only observations that are collocated geographically and temporally. The coincidence criteria used varies and is stated in each subsection below. Note that throughout these comparisons we use the AMSR-MODIS level 2 data (that is, we use the data before griding it), to allow a better comparison. In analyzing these comparisons, it is important to bear in mind that each of

the observations used is sampling different volumes; sondes are precise in-situ measurements which represents conditions at a local point, AMSR-MODIS level 2 product estimates the boundary layer conditions within a pixel size of 1 km at nadir, while GPS-RO samples through the limb of the atmosphere, averaging over large horizontal distances of ∼200 km. Hence, geophysical variability will inevitably complicate the interpretation of such comparisons.

### 3.1  Radiosondes

In the comparison shown here we used sondes from two field campaigns: (1) the Alfred Wegener Institute (AWI) Polarstern laboratory campaign with more than 50 expeditions to the Arctic and the Antarctic (König-Langlo and Marx, 1997) since



1982 and (2) the Marine Atmospheric Radiation Measurement (ARM) GPCI Investigation of Clouds (MAGIC) campaign with approximately 20 round trips between Los Angeles and Honolulu during 2012-2013 (Kalmus et al., 2014; Zhou et al., 2015).

To compute the BL-CWV from these measurements, we first identified the boundary layer inversion height and then integrated the specific humidity profile from that height to the surface. We use three different methods to find the inversion: the location of the minimum vertical gradient of specific humidity, the location of the minimum vertical gradient of relative humidity, and the location of the maximum vertical gradient of potential temperature. As in Millán et al. (2016), we exclude all the data below 200 m or above 4 km, and we use only robust inversions. That is, those inversions where the boundary layer inversion height estimates of the three methods agree within 200 m.

Figure 2-top shows the scatter between AMSR-MODIS and radiosonde BL-CWV within ±10 km and ±6 h. The best-fit line has a slope of 0.73, an RMS deviation of 0.50, and a correlation coefficient of 0.56, which suggests a reasonable but imperfect agreement between the two datasets. By decreasing the coincidence criteria distance from 10 to 1 km (Figure 2-bottom) it is possible to improve these metrics (the best-fit line slope becomes 0.75, the RMS deviation 0.39, and a the correlation coefficient of 0.71) but the total number of matches decreases from 307 to 124. Despite the scatter and the bias between the datasets, we find these results encouraging. The scatter was to be expected due the inherently noisy nature of the AMSR-MODIS product and because we do not know the extent to which the sonde measurements are representative of the average BL-CWV in the MODIS pixel.

## 3.2 GPS-RO

As cross-validation, we use GPS-RO data. This technique uses phase delays in the GPS signals collected from a receiver on board of a low Earth orbiting satellite to derive profiles of refractivity. From these profiles, humidity in the middle and lower troposphere can be derived. In particular we use GPS-RO data from the Constellation Observing System for Meteorology, Ionosphere, and Climate (COSMIC) constellation. A description of the measurements and the retrieval technique can be found in Kursinski et al. (1995), Kursinski and Hajj (2001), and Hajj et al. (2002). The accuracy of these measurements is around 10 to 20% below 7 km and 5% or better in the boundary layer (Kursinski et al., 1995). In particular we use version 2.6.

To compute the BL-CWV from GPS-RO we follow a similar methodology as in the AMSR-MODIS dataset. First, we match-up the GPS-RO measurements with AMSR. As coincidence criteria we assume a match when any GPS-RO lands within an AMSR footprint and ±6 hours. Then, following Ao et al. (2012), we identified the boundary layer inversion height as the minimum vertical gradient of the refractivity, which corresponds to the height where the refractivity changes most rapidly, and integrate the humidity profile from that height *upwards* to compute the CWV above the inversion height. Lastly, we subtract these estimates from the AMSR total CWV to compute the BL-CWV.

As an additional constraint we use the sharpness parameter, defined as the minimum refractivity gradient relative to the RMS value of the gradient averaged over the bottom 6 km of the atmosphere (see Ao et al. (2012) for more information), to identify regions where the BL inversion is well defined. As discussed by Ao et al. (2012), we found that the sharpness parameter is largest over the eastern subtropical oceans where stratocumulus occur (see Figure 3), with maximum average values of around



2.7 near the cost of Chile. The smallest sharpness parameters can be found in the ITCZ where the boundary layer is not well defined.

Figure 4-top shows the scatter between AMSR-MODIS and GPS-RO BL-CWV using as coincidence criteria $\pm 10\,\mathrm{km}$ and $\pm 6\,\mathrm{h}$ and a sharpness parameter value greater than 2.5. Again, despite a fair amount of scatter and bias, the degree of agreement

between the two datasets lends confidence in the usefulness of the AMSR-MODIS BL-CWV. By increasing the sharpness parameter requirement from 2.5 to 3.0 (Figure 4-bottom) the relationship between these two datasets improves with the best-fit line slope becoming 0.71, the RMS deviation 0.57, and the correlation coefficient 0.54. However, the total number of matches decreases from $\sim 23500$ to $\sim 750$. This improvement arises because when using a larger sharpness parameter we are ensuring that most pairings are in the stratus regions where the AMSR-MODIS technique should work better.

Through these comparisons, a consistent picture emerges suggesting either an underestimation of the AMSR-MODIS BL-CWV or an overestimation of the radiosonde and GPS-RO BL-CWV. An underestimation of the AMSR-MODIS BL-CWV has two possible reasons, an underestimation of the total CWV by AMSR and/or an overestimation of the MODIS CWV above the clouds. We found an excellent agreement between the AMSR total CWV versus the radiosondes measurements (not shown), with a strong correlation coefficient (0.94), a best-fit line slope of 1.06 and an RMS deviation of 0.28 cm. This suggest that

there may be an overestimation of the MODIS CWV above the clouds. The retrieval of BL-CWV above clouds is complicated by the fact that the near IR radiation penetrates the cloud layer. The multiple scattering of the light within the cloud increases the optical path length of the cloud and should result in an overestimate in water vapor above the clouds. The MODIS algorithm does not account for this effect and as a result the cloudy pixels are flagged with marginal quality assurance.

We believe that a consistent overestimation of the radiosonde and GPS-RO BL-CWV is unlikely due to the sharp gradients

associated with the boundary layer inversion but we do suspect that uncertainties in determining such inversion are one likely culprit causing some of the scatter shown in Figures 2 and 4. In some cases, it is difficult to determine the boundary layer inversion height in the radiosonde and in the GPS-RO data because several alternating dry and moist layers may be present in the measurements. In those cases, there is no guarantee that the algorithms chosen will identify the correct height, choosing instead a residual layer or a dry intrusion, which will lead to an overestimation or underestimation, respectively, of the BL-

CWV estimated by the radiosondes or GPS-RO data. von Engeln and Teixeira (2013) have shown that using different methods to estimate the boundary layer inversion height can lead to significantly different results even when using the same original datasets. For example, a consistent overestimation of the boundary layer inversion height (at least in the radiosonde cases) might be possible because as shown by Seidel et al. (2010) finding the inversion using the location of the minimum (maximum) vertical gradient of relative humidity (potential temperature) consistently yield higher PBL height estimates than other methods.

Nevertheless considering the boundary layer geophysical variability (for example, the short response time of the boundary layer), the different sampling volumes associated with each technique, and the uncertainties in determining the boundary layer inversion height, we conclude that AMSR-MODIS BL-CWV, sondes, and GPS-RO BL-CWV measurements are in good agreement.





## 4  Results

### 4.1  Climatology of stratus amount

Figure 5 shows the total number of observations found throughout the AMSR-MODIS dataset from 2002 to 2017. High number of observations means that uniform liquid cloud fields were found consistently in such areas, and can be interpreted as a qualitative proxy for stratus cloud fraction amount. Overlaid on this map are contours displaying the mean vertical velocity at 500 hPa ($\omega_{500}$) from ERA-Interim (Dee et al., 2011) showing regions of large scale subsidence and convective regions. As expected, high number of observations are found in subtropical eastern oceans, in regions where stratocumulus clouds frequently occur (e.g. Klein and Hartmann, 1993; Wood, 2012). These subtropical regions are characterized by relatively cold sea surface temperature, strong subsidence, and well defined temperature inversions at the boundary layer (see for example, the high values of the sharpness parameter shown in Figure 3). High number of observations can also be found in regions where stratus clouds frequently occur (e.g. Teixeira, 1999) like over the arctic, over the southern ocean, and off east coast of the continents in the northern hemisphere. The lowest number of observations are found in the deep tropics, particularly in convective regions where the presence of non-uniform cumulus and also obscuring high clouds associated with deep convection decreases considerably the probability of finding uniform liquid cloud fields. Hence, the observations in this tropical region, where the boundary layer is not well defined, are not particularly reliable.

Climatological annual cycles of the number of observations for the regions shown in Figure 5 are shown in Figure 6. These regions are subtropical stratus locations taken from Klein and Hartmann (1993) and listed in table 1 for clarity. The annual cycles in the Californian and Canarian regions are similar with maxima during July and the peak lasting from June to August, however, the Canarian region has far fewer observations (i.e. unobscured stratus clouds). The annual cycle is notably stronger in the Peruvian and Namibian regions with maxima during August and the peak lasting from June to November. Overall, the annual cycle of the number of observations is in good qualitative agreement with the climatology of marine stratus compiled from ship-based weather observations by Klein and Hartmann (1993) or the climatology of low clouds derived from 5 years of CloudSat and CALIPSO data by Muhlbauer et al. (2014).

Previous studies have suggested that the seasonality of this type of clouds largely follows the lower tropospheric stability (LTS) (Klein and Hartmann, 1993; Richter, 2004; Wood and Bretherton, 2006; Richter and Mechoso, 2006). Figure 7 shows the annual cycle of LTS taken from the ERA-Interim reanalysis. LTS is defined as the difference between potential temperature at 700 hPa and the temperature at the surface. The LTS relation can be theoretically derived from the energy balance equation for the boundary layer (Chung et al., 2012) and can be thought of as a proxy for the strength of the inversion capping the boundary layer; in principle, a strong inversion is more effective at trapping humidity in the boundary layer, which will gradually accumulate and reach saturation, hence, enhancing cloud cover. As displayed, the Canarian LTS annual cycle is similar to the Californian one but ∼4°K lower throughout the year, which as suggested by Klein and Hartmann (1993) may result in the significantly reduction of stratus in such region. These LTS annual cycles are similar to the ones shown or described by Klein and Hartmann (1993). More interestingly, Figure 7 also shows the correlation coefficient between the number of observations and LTS in each of these regions. As expected, relatively high values can be found in most regions (0.77, 0.8, 0.91, and 0.93 for

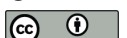



the Californian, Canarian, Namibian, and Peruvian regions respectively). Interannual correlations, that is, correlations based upon the monthly time series as opposed to the climatological data, also display relative high correlations, with values of 0.85, 0.87, 0.76, and 0.77 for the Peruvian, Namibian, Californian, and Canarian regions, respectively.

### 4.2 Climatology of BL-CWV

Figure 8 shows the annual cycle for BL-CWV, SST, and BL-CTH taken from the AMSR-MODIS dataset. Only the Peruvian and Namibian region display a significant BL-CWV annual cycle with a maximum to minimum differences of 8 and 6 mm, respectively; displaying a clear sinusoidal signature (specially in the Peruvian region) with maxima in February and minima during the fall. In the other regions, the maximum to minimum BL-CWV difference is only 3 mm throughout the year with no well defined minima or maxima. All regions display a clear SST annual cycle, with maximum to minimum differences close
to $\sim 4°$K. As with the LTS annual cycles shown in Figure 7, these SST annual cycles agree with the ones shown or described by Klein and Hartmann (1993). The BL-CTH annual cycles display a lot of variability, with no clear discernible pattern among the regions. The Canarian and Peruvian regions show the greatest maximum and minimum differences with 1.5 and 0.9 km respectively.

  Table 2 shows the climatological and interannual correlation coefficients between the BL-CWV annual cycle and the ones
found for BL-CTH, SST, LTS, and the number of observations. Only the Peruvian and Namibian regions display high correlation coefficient (that is, |r| > 0.7), at least in the climatological correlations, between these parameters. In those two regions the seasonal cycle strongly follows a cycle of modulation of the SST, which is negatively correlated with the LTS, and positively correlated with boundary layer depth, and bulk boundary layer water vapor content. This pattern is also true with weaker correlation in the Californian and Canarian regions which may be due to the smaller seasonal amplitude of the cycles in these
regions.

  Figure 9 shows the measured annual cycle for BL-CWV, as well as the derived one from a simple well-mixed boundary layer model as the one described by Millán et al. (2016), assuming a surface relative humidity of 80% (both normalized by their respective maximum values). The modeled BL-CWV does resemble the BL-CWV measured one, particularly in the Peruvian and Namibian regions, where the correlation coefficients between the modeled and measured BL-CWV are 0.99 and 0.95
respectively. This suggests than in the most robust subtropical stratocumulus regions key properties such as water vapor content can be represented by a simple mixed-layer model. Note, however, that the well-mixed model consistently overestimates the measured BL-CWV in part due to the underestimation of the AMSR-MODIS product as shown by Figure 2 and Figure 4.

### 4.3 Stratocumulus to Cumulus transitions

  To further analyze the data, we focused on typical Stratocumulus-Cumulus transects. In these transects, stratiform clouds
typically reside above relatively cold waters near the coasts, below subsiding air, in shallow and normally well mixed boundary layers capped by a strong temperature inversion. As trade winds advect air toward the equator, the subsidence weakens and the sea surface gradually warms leading to an increase in heat and moisture fluxes and a rising and weakening of the inversion, resulting in trade wind shallow convective clouds and eventually in deep convective clouds (e.g., Teixeira et al., 2011).



Figure 10 displays the transects used. These transects were taken from Sandu et al. (2010), in particular the ones constructed using gridded mean climatological meteorological fields. Figure 11 shows the climatological SST, BL-CWV and BL-CTH along these transects. The Californian and Canarian transects display data from June, July, and August while the Peruvian and Namibian transects for September, October, and November. These months correspond to the ones used by Sandu et al. (2010)

during their trajectory analysis. These are the periods where Klein and Hartmann (1993) found the highest cloud fraction in the stratocumulus region on each oceanic basin.

The Californian and Canarian, transects display the expected behavior with warmer temperatures towards the equator resulting in a systematic deepening and moistening of the boundary layer. The boundary layer cloud top height starts as shallow as 1.4 and deepens up to 2.4, or 2.5 km in the Californian and Canarian transects respectively. Similarly, the boundary layer

column water starts as dry as 7 or 11 and moistens up to 22 or 25 mm, respectively. On the other hand, the Namibian and Peruvian transects do not display this "canonical" picture. Notably these southern hemisphere transects each cross the equator. In the Namibian transect, despite a clear increase in SST along it, BL-CTH remains constant, at around 1.5 km, throughout its entire length. On the other hand, BL-CWV shows a systematic moistening, starting as dry as 7 and going as high as 20 mm. In the Peruvian transect, despite a clear increase in SST, BL-CTH and BL-CWV remains constant (with values of 1.9, km and

10 mm) up to 2500 km into the transect; only deepening and moistening steeply due to a sharp jump in the SSTs as the transect crosses the ITCZ.

## 5    Summary

The synergy of AMSR and MODIS measurements provides the opportunity of estimating for the first time the column of water vapor inside the marine boundary layer, although the technique is limited to homogeneous cloud fields during daylight.

The boundary layer water vapor information results from combining AMSR estimates of total column water vapor, which are unaffected by clouds, with those derived from MODIS near-infrared channels using solar radiation reflected by clouds, which estimate the water vapor above the clouds. In this study we discussed results from the second public release of the AMSR-MODIS dataset. That is, version 2.0, which only difference against version 1 is that high clouds are masked out using the cloud phase optical properties (only using clouds which phase have been identified as liquid by the cloud thermodynamic phase

classification algorithm. The AMSR-MODIS dataset is available in daily and monthly composites with a 1° by 1° resolution. Monthly files were constructed aggregating the daily files but disregarding daily pixels with standard deviation greater than 0.2 cm. This threshold mostly rejects pixels in the ITCZ where the boundary layer is not well defined. As a by product of the BL-CWV algorithm, the AMSR-MODIS dataset also provides the BL-CTH, BL-CTP, and the BL-CTT of the MODIS pixels used, as well as the associated SST and total CWV from AMSR. As such, the AMSR-MODIS dataset provides many of the

variables of interest for boundary layer studies.

We exploited 16 years of collocated AMSR and MODIS measurements to study the behavior of the number of observations as well as the behavior of the BL-CWV on well known stratus regions. Further, we also study the Sc-Cu transitions. The main findings can be summarized as follows:





- Comparisons between AMSR-MODIS BL-CWV against radiosondes and GPS-RO data were undertaken. A consistent picture emerges suggesting an underestimation of the AMSR-MODIS BL-CWV measurements most likely due to an overestimation by the water vapor column above the clouds by MODIS. However, considering the geophysical variability of the boundary layer, the different sampling volumes of each technique, as well as the uncertainties associated with determining the inversion height in the sondes and GPS-RO boundary layer estimates, we believe that the comparisons demonstrate the skill of the AMSR-MODIS boundary layer water vapor estimates to detect variability.

- In well know stratus regions, the annual cycle of the number of observations (a qualitative proxy for stratus cloud fraction amount) is in good qualitative agreement with the climatology of marine stratus compiled from ship-based weather observations by Klein and Hartmann (1993) and the climatology of low clouds derived from 5 years of CloudSat and CALIPSO data by Muhlbauer et al. (2014). Furthermore, as previous studies have suggested, in all the stratus regions the number of observations is well correlated with lower tropospheric stability showing the inclination of stratus (homogeneous clouds fields) to form under a strong capping inversion layer.

- In the most robust subtropical stratocumulus regions key properties such as water vapor content can be represented by a simple mixed-layer model.

- The Californian and Canarian stratocumulus to cumulus transitions displayed the "canonical" view of these transects with a gradual deepening and moistening of boundary layer as the sea surface temperature warm up towards the equator. On the other hand, the Namibian and Peruvian transects do not display this canonical behavior.

In summary, these results demonstrate that the AMSR-MODIS dataset provides useful information regarding the marine boundary layer, particularly over stratus regions. Further, the multi-sensor nature of the analysis demonstrates that there exists more information on boundary layer water vapor structure in the satellite observing system than is commonly assumed when considering the capabilities of single instruments.

*Data availability.* The AMSR-MODIS dataset can be found on the NASA Goddard Space Flight Center Earth Sciences (GES) Data and Information Services Center (DISC) website (http://disc.sci.gsfc.nasa.gov/) with "10.5067/MEASURES/AMDBLWV2" and "10.5067/MEASURES/AMMBLWV2" digital object identifiers for the daily and monthly data respectively. The data is stored in netcdf version 4 format. ERA-Interim reanalysis fields can be found at the ECMWF website (http://apps.ecmwf.int/datasets/).

*Author contributions.* Luis Millán wrote the AMSR-MODIS algorithm, and carried out the analyses. Matthew Lebsock and Joao Teixeira provided scientific expertise throughout all stages of the research.





*Acknowledgements.* The research described in this study was carried out by the Jet Propulsion Laboratory, California Institute of Technology, under contract with the National Aeronautics and Space Administration.



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



**Figure 1.** Example of daily (January 1st, 2005, left) and monthly (January 2005, right) composites of BL-CWV (top), its standard deviation (middle), and the number of observations used (bottom).

**Table 1.** Geographical extent of the regions used in this study.

| Region | Geographical boundaries |
|--------|------------------------|
| Peruvian | 10°-20° S, 80°-90° W |
| Namibian | 10°-20° S, 0°-10° E |
| Californian | 20°-30° N, 120°-130° W |
| Canarian | 15°-25° N, 25°-35° W |



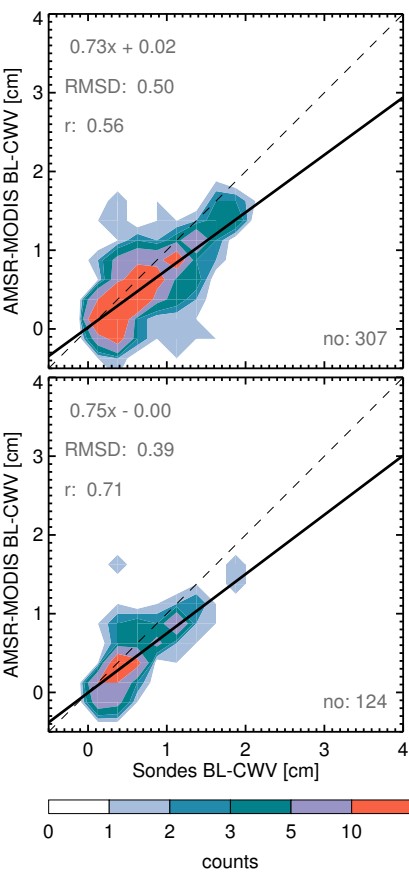

**Figure 2.** Sondes BL-CWV measurements scattered against the AMSR-MODIS BL-CWV estimates using $\pm 10$ km and $\pm 6$ h (top) and $\pm 1$ km and $\pm 6$ h (bottom) as coincidence criteria. The dashed black line is the one-to-one line. The solid black line displays a linear fit. The root mean square deviation, the linear fit equation, the correlation coefficient R, and the total number of matches are shown.



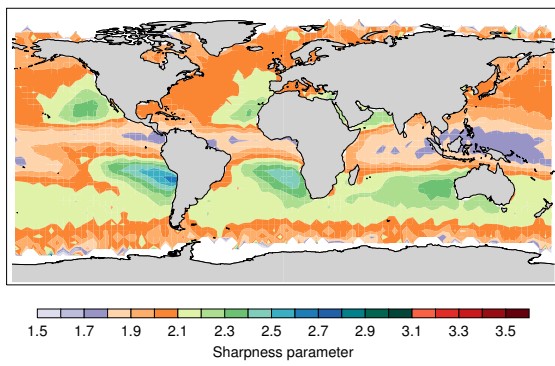

**Figure 3.** Sharpness parameter (relative minimum refractivity gradient) from 9 years (2006-2014) of the COSMIC data used on a 4° by 4° grid.

**Table 2.** Climatological (top) and interannual (bottom) correlation coefficients between BL-CWV and several other variables. Bold text indicates a high correlation coefficient (|r|>0.7)

| Region | SST | LTS | BL-CTH | Number of Observations |
|---|---|---|---|---|
| Peruvian | **0.95** | **-0.95** | **0.96** | **-0.98** |
| Namibian | **0.81** | **-0.76** | **0.81** | **-0.72** |
| Californian | 0.37 | -0.27 | 0.48 | -0.36 |
| Canarian | 0.06 | -0.55 | **0.72** | -0.26 |
| Peruvian | **0.95** | **-0.88** | **0.86** | **-0.92** |
| Namibian | **0.75** | -0.63 | **0.82** | -0.61 |
| Californian | 0.44 | -0.31 | 0.63 | -0.27 |
| Canarian | 0.12 | -0.22 | 0.61 | -0.10 |



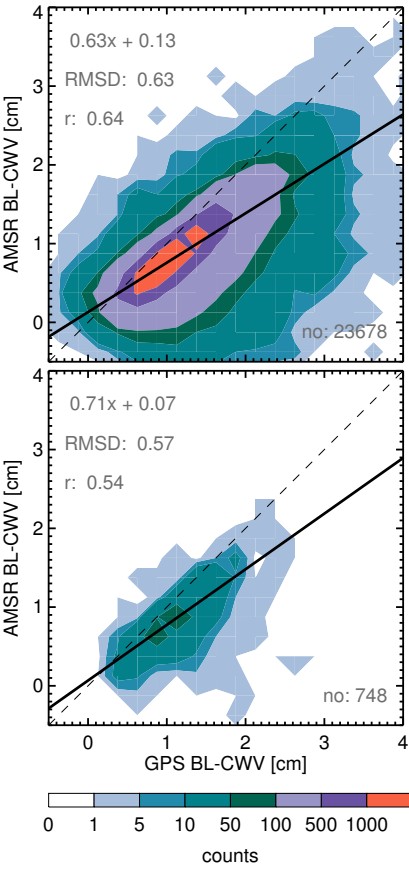

**Figure 4.** GPS-RO BL-CWV measurements scattered against the AMSR-MODIS BL-CWV measurements using $\pm 10$ km, $\pm 6$ h, and a sharpness parameter greater than 2.5 (top) and $\pm 10$ km, $\pm 6$ h, and a sharpness parameter greater than 3 (bottom) as coincidence criteria. The dashed black line is the one-to-one line. The solid black line displays a linear fit. The root mean square deviation, the linear fit equation, the correlation coefficient R, and the total number of matches are shown.




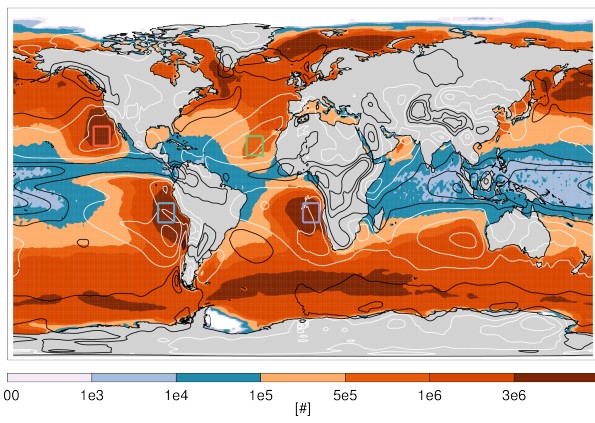

**Figure 5.** Number of observations found in the AMSR-MODIS dataset over 2002 to 2017. Overlaid contours display air vertical velocity at 500 hPa ($\omega_{500}$) from ERA-Interim, with white contours at 0.01,0.03, 0.05 Pa s$^{-1}$ denoting sinking of air and black contours -0.05,-0.03,-0.01 Pa s$^{-1}$ denoting rising of air. A 2D smoothing has been applied to the $\omega_{500}$ fields. Color rectangular boxes identify regions with high amount of stratocumulus clouds. These locations are adopted from Klein and Hartmann (1993).

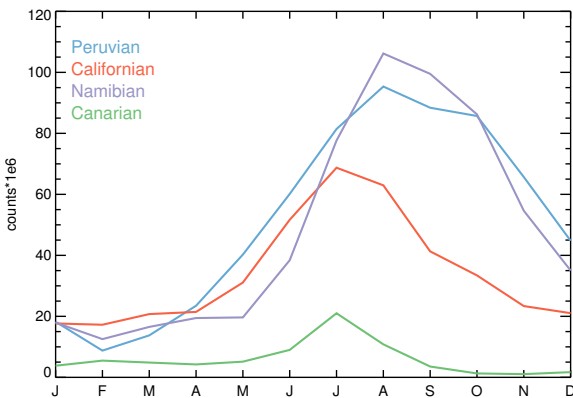

**Figure 6.** Annual cycle of the total AMSR-MODIS number of observations for the regions delimited in Figure 5 by the rectangular boxes.




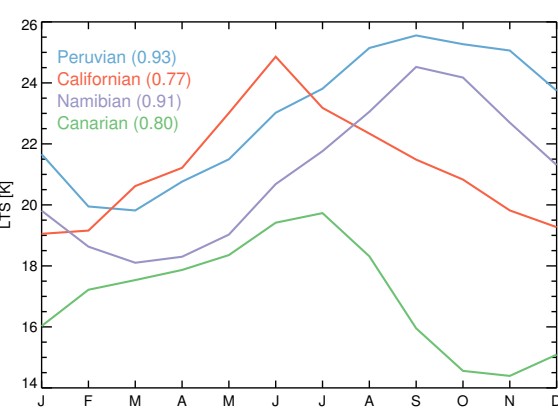

**Figure 7.** Annual cycle of LTS for the regions delimited in Figure 5 by the rectangular boxes. The number in brackets are the correlation coefficient between the annual cycle of the number of observations (shown in Figure 6) and these LTS cycles.





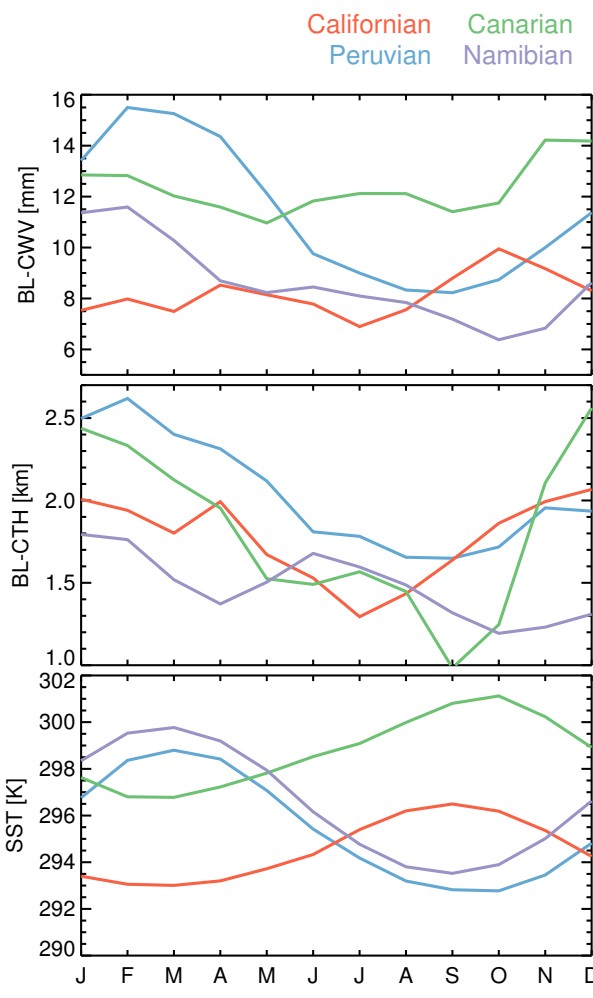

**Figure 8.** Seasonal cycle of BL-CWV, BL-CTH, and SST, for the regions delimited in the Figure 5 by the rectangular boxes.



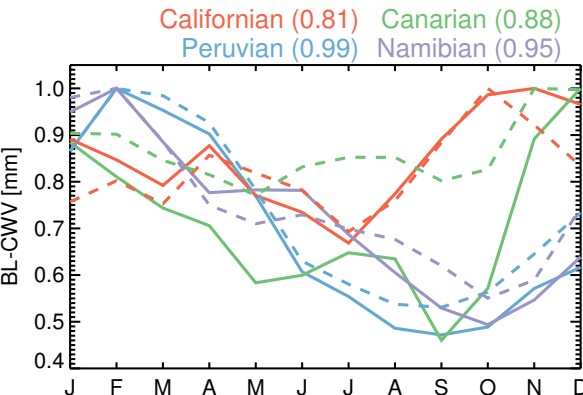

**Figure 9.** Normalize measured seasonal cycle of BL-CWV (solid lines), as well as derived from simple mixed layer model (dash lines) for the regions delimited in the Figure 5 by the rectangular boxes. The number in brackets are the correlation coefficient between the measured BL-CWV and the modeled one.

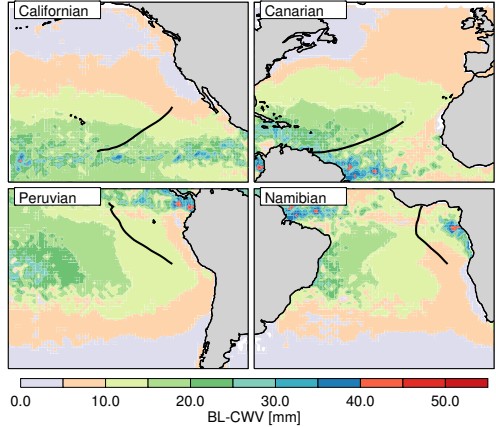

**Figure 10.** Transects along the climatological streamlines used in this study (taken from Sandu et al. (2010)). The contours show the climatological composite for all the AMSR-MODIS BL-CWV data available.





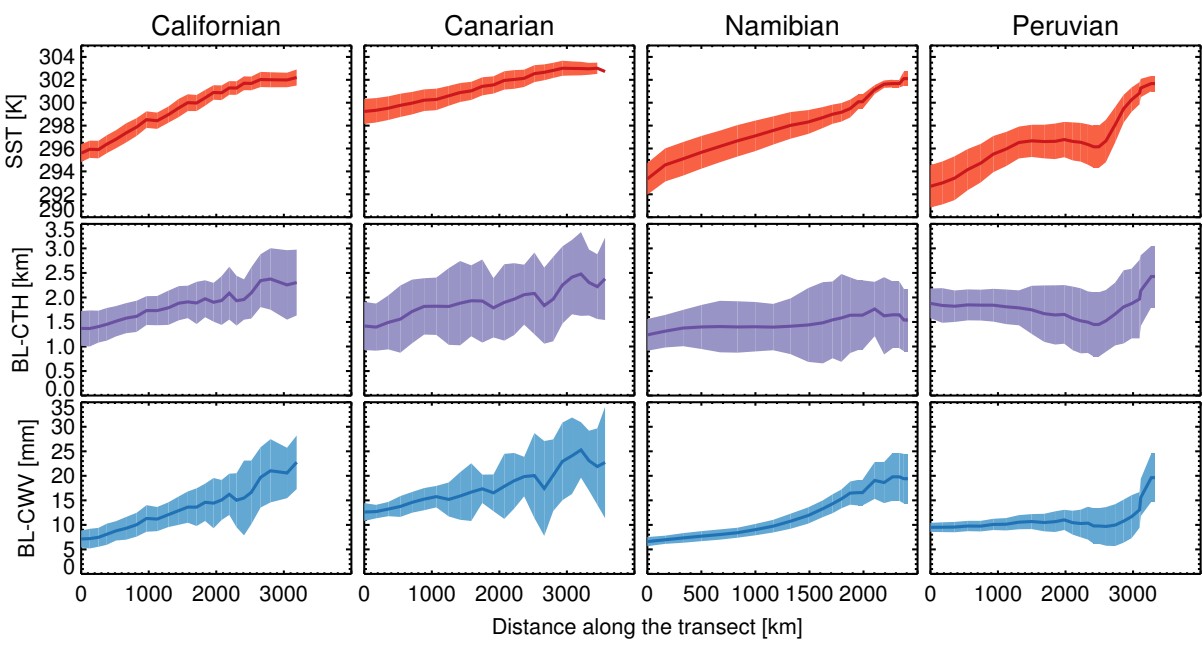

**Figure 11.** Climatological SST, BL-CTH and BL-CWV along the transects shown in Figure 10. The envelopes display the standard deviation.