# Peer review of "Variability of Bulk Water Vapor Content in the Marine Cloudy Boundary Layers from Microwave and Near-Infrared Imagery"

_Atmospheric Chemistry and Physics, 2019_

## Referee Comment (RC1) · Anonymous Referee #1 · 19 Mar 2019

General:

An interesting contribution on the synergetic use of different satellite measurements to monitor some PBL characteristics. I have a few major concerns, and various minor questions comments that I'd like to see addressed in an update. I rated the manuscript as Major Revision, mostly to leave the authors more time if any of my comments require larger work.

Major Issues:

- I am missing some more information on the systematically different sonde data vs. the RO data. Figure 2 and 3 seem to show different conditions, thus potentially sampling

different areas, with RO data having CWV data up to 3cm, while the sonde data extends only up to about 1.5cm. Is the RO sharpness primarily identifying regions that are more humid while the sonde data is in less humid areas? Is there any chance to also compare RO to sonde to improve the understanding? Or to include more data?

- Page 2, Line 31 (P2/L31): points to systematic issues in the December months. Just "removing" a month because it does not seem to fit, and use a another data version that fits, would need a much more substantiated justification. Thus I'd like to see more info on what might cause this issue and why the version 6.1 was used now. And then, why 6.1 is not used throughout.

Minor / Editorial Issues:

- Section 2: would it be worthwhile to point out that MODIS measures around 1:30pm?

- P2/L27/L33: the estimated errors are mentioned here, could you include also whether this is a systematic or random uncertainty / error?

- P3/L11: "as the number of the number": correct?

- P4/L7: "data below 200m or above 4km": I could of course look at the cited article, but maybe it can be better explained here. Does this mean if the inversion layer is below/above, the sonde is not used? Or is this data below 200m not used in the integral (which would be more of an issue)?

- P4/L8: How much data is removed in these screening steps? And did you try to loosen the threshold of the three methods to be within 200m to increase the sample size?

- P4/L24: The RO processing articles seem rather outdated, I assume that the processing uses some more recent algorithms, e.g. 1DVar (I might be wrong though). And where does the data come from, UCAR? The version 2.6 though is not a UCAR identifier as far as I can see. It would also be interesting to get more info on the humidity background if a 1DVar was used. Otherwise, RO will not provide very accurate info

on humidity in the mid troposphere.

- P6/26: When using ERA-I data, is exactly the same data at the same time used, or is this a larger data set?

- Figure 1: Please use different color range for BL-CWV and its std dev, to allow more visualization of the std dev values.

- Table 2: What is the last column? It seems not the number of obs.

- Figure 4: There appear to be also negative CWV in this plot, is that found in certain regions/for low CWV values? Any reason why this is not excluded from the data set? Seems to point to the MODIS overestimation you mentioned.

- Figure 5: These rectangular/boxes are from a publication in 1993. Some question on that: Why is the Australian not included? Why not use all identified regions in that article(or at least the mid-lat marine stratus ones)? And last, is there a good reason to update the boxes with the latest data available? We have a better, higher resolution picture of our planet compared to 1993.

- Figure 6, 7, 8, 9: my color print out shows almost exactly the same colors for Peruvian and Namibian. I cannot distinguish them at least. On screen it is okay. We do have high end color printers, thus I assume this might also happen for others printing it out. When zooming into Figure 5, I noted that the boxes have the same color, but I think that is not really necessary and limits your color options.

- Figure 6: Maybe I missed it somewhere in the discussion, but this has no NH/SH shift, but that is found in Figure 7 and others. In 6, it seems they all peak around July/August.

- Figure 6: Is the result better visible if it is normalized to the total in the area? Also, these regions are different in size and cover different subsidence, does that have any impact?

- Generally, is there any impact visible of the different AMSR instruments used here?

---

## Author Comment (AC1) · 29 Apr 2019

**Response to Anonymous Referee #1**

We greatly thank the reviewer for his/her comments. Below are our responses in blue

Reviewer comments

General: An interesting contribution on the synergetic use of different satellite measurements to monitor some PBL characteristics. I have a few major concerns, and various minor questions comments that I'd like to see addressed in an update. I rated the manuscript as Major Revision, mostly to leave the authors more time if any of my comments require larger work.

Major Issues:

- I am missing some more information on the systematically different sonde data vs. the RO data. Figure 2 and 3 seem to show different conditions, thus potentially sampling different areas, with RO data having CWV data up to 3cm, while the sonde data extends only up to about 1.5cm. Is the RO sharpness primarily identifying regions that are more humid while the sonde data is in less humid areas?

As the reviewer points out, the difference is mainly sampling different areas, that is, when the sharpness parameter is greater than 2.5 (top panel in figure 4) the comparison encompasses regions where a lot of cumulus clouds might be present (see Figure 3 for reference), hence, the BL-CWV is higher. However, when the comparison is made for sharpness parameters greater than 3 (bottom panel in figure 4) the comparison is mostly over stratus regions and the BL-CWV values resemble more the sonde values. The sondes on the other hand are always restricted to stratus regions since we only used robust inversions, that is, when the three different methods (explained in the text) agree within 200m.

The following sentences will be modified,
(P4 line 7 of the original manuscript): That is, those inversions where the boundary layer inversion height estimates of the tree methods agree within 200m, **which mostly occur in stratus regions**.

(P5 line 8 of the original manuscript): This improvement arises because when using a larger sharpness parameter, we are ensuring that most pairings are in the stratus regions **(see Figure 3 for reference)** where the AMSR-MODIS technique should work better. **A larger sharpness parameter also reduces the range of the BL-CWV comparison by excluding the high values found under cumulus regimes. This makes the comparison ranges (that is, in Figure 4-bottom) similar to the ones found in the sonde comparison where the sondes used are restricted to stratus regions by using the robust inversion criteria. That is, when the three different methods to find the inversion (explained in section 3.1) agree within 200m.**

Is there any chance to also compare RO to sonde to improve the understanding? After careful consideration, we feel that such comparison belongs to a GPS-RO validation paper and not here. However, we hope that our previous explanation will suffice for the reviewer.

Or to include more data?
We included as much radiosonde and GPS-RO data as we could find.

- Page 2, Line 31 (P2/L31): points to systematic issues in the December months. Just "removing" a month because it does not seem to fit, and use a another data version that fits, would need a much more substantiated justification. Thus I'd like to see more info on what might cause this issue and why the version 6.1 was used now. And then, why 6.1 is not used throughout.

The issue was a one-off coding error on the MODIS processing algorithm (per the personal communication with Richard Frey). We discover the issue after the whole dataset was produced with version 6.0 and a reprocess of the AMSR-MODIS using version 6.1 through-out the entire time period is currently outside our possibilities due to the large time involved in downloading the terabytes of MODIS data.  We will modify the following sentence: Instead, version 6.1 was used for all December months **as recommended by the MODIS team.  A full reprocessing of the AMSR-MODIS dataset using MODIS version 6.1 (or the latest MODIS version) is left for a future AMSR-MODIS version.**

Minor / Editorial Issues:

- Section 2: would it be worthwhile to point out that MODIS measures around 1:30pm? The following sentence will be modified to: All these instruments orbit in tandem measuring the same volume of air within minutes of each other, that is, by design, these measurements are collocated; their equatorial crossing time is ~1:30pm.

- P2/L27/L33: the estimated errors are mentioned here, could you include also whether this is a systematic or random uncertainty / error? Both of these estimates are random errors. This will be reflected in the text.

- P3/L11: "as the number of the number": correct?  The sentence will be changed to:  It also shows its associated standard deviation as well as the number of single observations (MODIS pixels) …

- P4/L7: "data below 200m or above 4km": I could of course look at the cited article, but maybe it can be better explained here. Does this mean if the inversion layer is below/above, the sonde is not used? Or is this data below 200m not used in the integral (which would be more of an issue)?
We only exclude this data in the BL height determination analysis, the sentence will be modified to: As in Millán et al. (2016), **during the inversion height determination**, we exclude all the data below 200 m or above 4 km **to avoid artifacts caused by temperature inversions near the surface as well as to avoid free-tropospheric features**. **Further**, we use only robust inversions, that is, …

- P4/L8: How much data is removed in these screening steps? And did you try to loosen the threshold of the three methods to be within 200m to increase the sample size?
We did not quantify how much data was screen by those steps, because such a stringent threshold is needed to ensure that the radiosondes used represent a well behave boundary layer profile. As shown in the following figure, relaxing such threshold could result in using sonde profiles where the boundary layer is not well defined.

[Figure]

Examples of relative humidity (RH) sonde measurements: (left) a robust inversion and (right) an unrobust inversion. Color lines display the boundary layer height determined using the location of the minimum vertical gradient of RH (red), the location of the minimum vertical gradient of specific humidity $q$ (blue), and the location of the maximum vertical gradient of potential temperature Θ (green). Figure 3 from Millán et al (2016) doi: 10.1175/JAMC-D-15-0143.1

- P4/L24: The RO processing articles seem rather outdated, I assume that the processing uses some more recent algorithms, e.g. 1DVar (I might be wrong though). And where does the data come from, UCAR? The version 2.6 though is not a UCAR identifier as far as I can see. It would also be interesting to get more info on the humidity background if a 1DVar was used. Otherwise, RO will not provide very accurate info on humidity in the mid troposphere. The data comes from the JPL retrieval algorithm, for which those are the most current references. The following sentence will be modified to: In particular we use version 2.6 **of the JPL processing algorithm.**

- P6/26: When using ERA-I data, is exactly the same data at the same time used, or is this a larger data set? The following sentence will be added at the end of that paragraph: Note that for each region, ERA-Interim data from the nearest synoptic time (0,6,12,18 UT) to the measurement local time was used.

- Figure 1: Please use different color range for BL-CWV and its std dev, to allow more visualization of the std dev values. After careful consideration, we decided that it was best to leave the colorbar as it is. This way, the reader gets an immediate sense of the size of the standard deviation, as opposed to have to look at the ranges of two colorbars.

- Table 2: What is the last column? It seems not the number of obs. It is the correlation between BL-CWV and the number of observations. The caption of the table will be change to: Climatological (top) and interannual (bottom) correlation coefficients between BL-CWV and **SST, LTS BL-CTH and the number of observations**.

- Figure 4: There appear to be also negative CWV in this plot, is that found in certain regions/for low CWV values? Yes and no, that is, they do tend to occur more at low CWV values but not only at those places. Any reason why this is not excluded from the data set? Seems to point to the MODIS overestimation you mentioned.  We believe that excluding these points could lead to a high bias of the BL-CWC. The following sentence will be added after the MODIS overestimation discussion:  The overestimation of the MODIS CWV above the clouds could lead to negative values in the AMSR-MODIS dataset (as can be seen in Figure 4). However, we do not recommend that these negatives values are excluded of any analysis of the AMSR-MODIS dataset because some negative values will be due to the noisy nature of the MODIS measurements over cloudy pixels, and excluding those will lead to biasing high.

- Figure 5: These rectangular/boxes are from a publication in 1993. Some question on that: Why is the Australian not included? The Australian region is a relatively weaker SC region and hence is harder to interpret. Why not use all identified regions in that article(or at least the mid-lat marine stratus ones)? Again, because we focus in the robust SC regions. And last, is there a good reason to update the boxes with the latest data available? We have a better, higher resolution picture of our planet compared to 1993. Several studies have used these boxes and we wanted to compare against them without adding an extra level of complexity by changing the study regions, but the reviewer is correct, and a paper studying the impact of choosing different regions in the SC vicinities and its impact upon stratus amount and BL-CWV would be interesting.

- Figure 6, 7, 8, 9: my color print out shows almost exactly the same colors for Peruvian and Namibian. I cannot distinguish them at least. On screen it is okay. We do have high end color printers, thus I assume this might also happen for others printing it out. When zooming into Figure 5, I noted that the boxes have the same color, but I think that is not really necessary and limits your color options. The color for the Namibian region will be changed for a darker purple.

- Figure 6: Maybe I missed it somewhere in the discussion, but this has no NH/SH shift, but that is found in Figure 7 and others. In 6, it seems they all peak around July/August.  The difference is not as visible in Figure 6 as it is in Figure 7, however, it is there. As stated in the text (P6 line 19 of the first version): The annual cycle us notably stronger in the Peruvian and Namibian regions with maxima during August *(as opposed to July)* and the peak lasting from June to November.

- Figure 6: Is the result better visible if it is normalized to the total in the area?
Below is the figure as requested by the reviewer, as shown is really similar to the Figure shown in the original paper. The reason is because the area of the regions is really similar (all regions shown are 10 by 10 degrees) which is roughly 1e6 KM2 (the number we used originally to scale them.  In responding to this question, we realize that the y-label was wrongly shown as counts*1e6 when it should have been counts / 1e6. This will be updated in the new manuscript.

[Figure]

Figure: Annual cycle of the total AMSR-MODIS number observations (scaled by the total area of each region) for the regions delimited in figure 5 by the rectangular boxes

Also, these regions are different in size and cover different subsidence, does that have any impact?
As mention above, they are really similar in size, and while they do have a somehow different subsidence (see figure below) , the counts are not consistently correlated with subsidence (in this case using omega500 from ERA-Interim),  as LTS is (as shown in Figure 7 of the manuscript).

[Figure]

Figure: Annual cycle of w500 (from ERA-Interim) for the regions delimited in figure 5. The number of brackets are the correlation coefficient between the annual cycle of the number of observations and these w500 cycles.

We will add in the manuscript: Other parameters (MODIS CTP, AMSR SST, ERA-Interim w500 and ERA-Interim Surface Pressure) were analyzed in a similar manner but none of them were strongly correlated with the number of observations across the four regions used here.

- Generally, is there any impact visible of the different AMSR instruments used here? We didn't find any, that is, we analyze time series of the AMSR total CWV and SST globally as well as in the regions study and did not find any visible impact / discrepancies. The following sentence will be added in the paragraph describing the AMSR instruments: Note that, no discrepancies nor visible impacts were found in time series from these two instruments.

---

## Referee Comment (RC2) · Anonymous Referee #2 · 2 May 2019

**Review for "Variability of Bulk Water Vapor Content in the Marine Cloudy Boundary Layers from Microwave and Near-Infrared Imagery" by Millán et al, submitted to Atmospheric Chemistry and Physics**

**General comments:**

This paper presents an assessment of boundary layer water vapor from satellite data. The method applied is based on a 16-year dataset of collocated near-infrared and microwave satellite observations.

In general, the paper is well structured, and provides some new interesting results. However, it needs some minor revisions before it can be published.

**Specific comments:**

p.2, l.4-5: you should mention that the datasets are derived from **satellite** observations

p.2, l.30-31: Other months do not show this inconsistency? What are the reasons for that?

p.2, l.34-35: Is this error (between 5 and 10 %) the error of the near infrared channels? Or the error of CWV? What about the error between cloudy and cloud-free cases? Is there a dependency on solar zenith angle?

p.3, l.6-7: Do you mean that you use only clouds that have been classified as "only liquid"? It is not clear here how you deal with mixed-phase clouds. The whole sentence should be rephrased for better clarity.

p.3, l.12-13: The monthly standard deviation of BL-CWV depends strongly on the variability of the boundary layer height (CTH). Have you checked this dependence?

p.3, l.31: Did you only use Arctic/Antarctic radiosondes? Which latitude belts did you include?

p.4, l. 10ff (and Fig. 2): I think that the variability within 6 hours is much larger than over 10 km. I guess most of the uncertainty reduction in the 1 hour/1km analysis comes from the shorter time range. Have you tried to keep 10 km (or even more) and reduce the temporal distance to 1 hour? In addition, 1 km drift of radiosondes is easily reached already within the boundary layer, therefore, I would suggest to neglect this "strong" 1 km criterion and rather focus on the temporal matching.

p.4, l. 20-23: It is known that GPS-RO data are missing some lower level inversions (especially below 1000 m above ground). How do you deal with this fact? Does it introduce a bias in your comparison?

p.5, l.3 (and Fig. 4): It is a bit misleading that you call the algorithms "AMSR-MODIS" and "GPS-RO". This suggests that the GPS-RO algorithm is independent, however you use GPS-derived CWV above the inversion and then subtract it from AMSR total column. Therefore, you are not comparing independent data here. Please comment on that!

p.5, l.7: Although slope and RMS decrease, the correlation coefficient also decreases. Do you have an explanation for that?

p.5, l.10-18: Is it possible that different viewing geometries or high solar zenith angles play a role in the uncertainties? If so, did you make separate analyses for different solar geometries or for different regions of the Earth?

p.6, l. 33-34 (and Fig. 7): What is the reason for the lower LTS in the Canarian region? Is it due to frequent advection of unstable air masses from the Saharan desert?

p.7, l.3: Why did you reverse the order of the regions here (compared to p.7, l.1)?

p.7, l.21-25: What are the model constraints? Vertical temperature structure? CWV? CTH?

p.7, l.26-27: I cannot see an overestimation since you are plotting normalized values in Figure 9. It would be good to see absolute values from the model! Does the magnitude of the overestimation is in line with the findings in Figures 2 and 4?

p.8, l.18: You are mentioning only here the restrictions of your method to homogeneous cloud fields during daylight. Does that affect the overall validity of your results? Do you expect a diurnal cycle?

p.8, l.23-25: This sentence (That is version2.0 (…) algorithm) is not necessary in the summary.

Figure 8:  Please provide information on the monthly variation of BL-CWV and BL-CTH, e.g. showing error bars or box-and-whisker plots

Figure 9: Since you plot normalized values, the unit [mm] is not correct!

**Technical corrections:**

p.1, l.20: replace "are" by "is"

p.1, l.24: "process**es**"

p.3, l.22: "criterion", not criteria

p.3, l.24: "grid**d**ing"

p.3, l.25: "represent" (not "represents")

p.5, l.1: "coast" (not "cost")

p.6, l.31, p.7, l.10: "4 K" (without degree sign)

p.8, l.14: "remain", not "remains"

p.8, l.32: "over", not "on"

p.8, l.32: "Sc-Cu": You never introduced these acronyms

Fig. 9 (caption): please correct: Normalize**d** (…) The number**s** (…) coefficient**s** (…)

---

## Author Comment (AC2) · 17 May 2019

**Response to Anonymous Referee #2**

We greatly thank the reviewer for his/her comments. Below are our responses in blue

**General comments:**
This paper presents an assessment of boundary layer water vapor from satellite data. The method applied is based on a 16-year dataset of collocated near-infrared and microwave satellite observations.

In general, the paper is well structured, and provides some new interesting results. However, it needs some minor revisions before it can be published.

**Specific comments:**
p.2, l.4-5: you should mention that the datasets are derived from **satellite** observations
That sentence will be changed to: The aim of this study is to show results from a ~16 year boundary layer column water vapor (BL-CWV) dataset derived from the synergy of microwave and near-infrared **satellite** imagery.

p.2, l.30-31: Other months do not show this inconsistency? What are the reasons for that?
As explained to reviewer 1, the issue was a one-off coding error on the MODIS processing algorithm (per the personal communication with Richard Frey). We will modify the following sentence: Instead, version 6.1 was used for all December months **as recommended by the MODIS team. A full reprocessing of the AMSR-MODIS dataset using MODIS version 6.1 (or the latest MODIS version) is left for a future AMSR-MODIS version.**

p.2, l.34-35: Is this error (between 5 and 10 %) the error of the near infrared channels? Or the error of CWV? What about the error between cloudy and cloud-free cases? Is there a dependency on solar zenith angle?
This is the error of the CWV, the sentence will be changed to: In particular, we use the CWV estimated using near-infrared channels. These CWV values have an estimated random error between 5% and 10% [Gao et al 2013].

There is no literature describing any differences between cloudy and clear-sky cases nor any dependence for solar zenith angle for the near IR CWV product. We will add the following sentence which will follow immediately after the Gao et al 2013 citation: These errors may have a solar zenith angle dependence as found for other MODIS products [i.e., Horvath et al (2013) , Grosvenor et al (2014)] and may worsen under cloud conditions, as such, we assume the 10% error through-out.

p.3, l.6-7: Do you mean that you use only clouds that have been classified as "only liquid"? It is not clear here how you deal with mixed-phase clouds. The whole sentence should be rephrased for better clarity.
The sentence will be changed to: We only use the clouds which have been classified, by the cloud thermodynamic phase classification algorithm (Plattnick et al 2015), as liquid. This is a completely re-written algorithm which instead of using a linear sequential structure, as in version 5, uses a voting discrimination logic to identify the cloud thermodynamic phase as ice, liquid or undetermined (Marchant et al. 2016).

p.3, l.12-13: The monthly standard deviation of BL-CWV depends strongly on the variability of the boundary layer height (CTH). Have you checked this dependence?

Yes, there is a strong dependence between CTH and the BL-CWV as expected. In a future study we will exploit its dependence to explore different bulk BL-CWV characterization (i.e. Stephens 1990- cropped at the CTH, a well-mixed model, a piecewise model, etc).

p.3, l.31: Did you only use Arctic/Antarctic radiosondes? Which latitude belts did you include?
The following figure will be added:

[Figure]

Caption: Map showing the geolocations of the radiosondes used in this study. Blue dots display the radiosondes that fulfill the criteria used in Figure 3-top while red dots display the subset that fulfill the criteria of Figure 3-bottom.

p.4, l. 10ff (and Fig. 2): I think that the variability within 6 hours is much larger than over 10 km. I guess most of the uncertainty reduction in the 1 hour/1km analysis comes from the shorter time range.
Note, that we do not have a 1hour/1km we display a 10km/6hours and a 1km/6hours.
Have you tried to keep 10 km (or even more) and reduce the temporal distance to 1 hour? In addition, 1 km drift of radiosondes is easily reached already within the boundary layer, therefore, I would suggest to neglect this "strong 1km" criterion and rather focus on temporal matching.

Below is a figure similar to figure 2 (in the original draft) keeping the 10km and reducing the temporal distance to 1 hour (bottom panel). As can be seen, these criteria result in only 34 matches and hence we prefer our previous one. Note that increasing the spatial threshold to 20km only results in 43 matches.

[Figure]

p.4, l. 20-23: It is known that GPS-RO data are missing some lower level inversions (especially below 1000 m above ground). How do you deal with this fact? Does it introduce a bias in your comparison?

As pointed by the reviewer in his/her next comment, we only use GPS above the inversion (when an inversion can be found) and subtract that estimate from the total CWV from AMSR.

p.5, l.3 (and Fig. 4): It is a bit misleading that you call the algorithms "AMSR-MODIS" and "GPS-RO". This suggests that the GPS-RO algorithm is independent, however you use GPS-derived CWV above the inversion and then subtract it from AMSR total column. Therefore, you are not comparing independent data here. Please comment on that!

We will change the section name to AMSR - GPSRO to avoid misleading the reader. Also, the first sentence will read: As cross-validation, we use **AMSR - GPSRO** data. The **GPSRO** technique uses phase delays ...

Also, we will add the following sentence: As such, a comparison between AMSR-MODIS and AMSR - GPSRO, is, in essence, a comparison between MODIS water vapor above the clouds and the GPSRO water vapor above the BL inversion layer.

p.5, l.7: Although slope and RMS decrease, the correlation coefficient also decreases. Do you have an explanation for that?

The RMSD depends on the values compared. If normalized, for example, by the mean of the AMSR-GPSRO values. The NRMSD are 0.50 and 0.49 for the sharpness parameter threshold of 2.5 and 3 respectively, that is to say, almost identical. We will update Figure 2 and 4 of the previous draft to use the NRMSD.

We will change the radiosonde comparison sentences to: The best-fit line has a slope of 0.73, a normalized (by the mean of the sondes values) root mean square deviation (NRMSD) of 0.69, and a correlation coefficient of 0.56 …
And to: By decreasing the coincidence criteria distance from 10 to 1 km (Figure 3-bottom) it is possible to improve these metrics (the best-fit line slope becomes 0.75, the NRMSD 0.59, and the correlation coefficient 0.71) but the total number of matches decreases from 307 to 124.

We will also change the GPSRO text to: By increasing the sharpness parameter requirement from 2.5 to 3.0 (Figure 5-bottom) the relationship between these two datasets improves with the best-fit line slope becoming 0.71 and the correlation coefficient 0.54. The NRMSD (in this case normalized by the mean of the AMSR-GPSRO values) remains nearly identical at ~0.5. However, the total number of matches decreases from ~23500 to ~750.

p.5, l.10-18: Is it possible that different viewing geometries or high solar zenith angles play a role in the uncertainties? If so, did you make separate analyses for different solar geometries or for different regions of the Earth?

As shown in the figures below (using a sharpness parameter of 2.5), there is some variation in the agreement between the two datasets per region, but not high enough to strongly indicate a viewing geometry/geolocation bias (at least in the stratus regions where most of the AMSR-MODIS observations are located, i.e. Figure 5 of the original draft).

[Figure]

[Figure]

p.6, l. 33-34 (and Fig. 7): What is the reason for the lower LTS in the Canarian region? Is it due to frequent advection of unstable air masses from the Saharan desert?

As discussed by Klein and Hartmann (1993), the SST for the Canarian region are about 3 to 5 degrees warmer than in the California region (this can be seen in figure 8 of the original manuscript) while the 700mb temperatures are really similar which results in a lower LTS.

p.7, l.3: Why did you reverse the order of the regions here (compared to p.7, l.1)?  We will change the order to be the same as in p7 l.1.

p.7, l.21-25: What are the model constraints? Vertical temperature structure? CWV? CTH?

The model description will be changed to:
Figure 10 shows the measured annual cycle for BL-CWV, as well as the derived one from a simple well-mixed boundary layer model as the one described by Millan et al (2016), assuming a surface relative humidity of 80\% **and using the AMSR SST temperature, the MODIS CTH and ERA-Interim surface pressure as constraints**.  These cycles were both normalized by their respective maximum values.

p.7, l.26-27: I cannot see an overestimation since you are plotting normalized values in Figure 9. It would be good to see absolute values from the model! Does the magnitude of the overestimation is in line with the findings in Figures 2 and 4?

The overestimation is not entirely explained by it, another reason is because the well mixed layer model is a oversimplification of the BL water profile, and it normally overestimates the BL-CWV. The following will be added: **and in part because of the simplistic representation of the boundary layer humidity profile by such a model.**

After consideration, we decided to show the figure with the normalized version to highlight that the water vapor observations can be, in general, be interpreted by a simple model as opposed to highlight the deficiencies of the model.

p.8, l.18: You are mentioning only here the restrictions of your method to homogeneous cloud fields during daylight. Does that affect the overall validity of your results? Do you expect a diurnal cycle?

The need of homogeneous cloud fields is mention in p3 l4: That is, we aim to identify homogenous fields of clouds in the MODIS data. Further, it is also mention, in section 4.1: High number of observations means that uniform liquid cloud fields were found consistently in such areas, and can be interpreted as …

The daylight restriction is implicit in P2 l5,6: Near infrared imagery provides the water vapor above the clouds (by measuring the solar radiation reflected near the 0.94-um water vapor band) while microwave imagery …

These restrictions do not affect our result because in regions where the boundary layer is well defined, the clouds tend to be homogenous, i.e. stratus clouds.

Although there is a boundary layer diurnal cycle, this does not affect our results, the AMSR-MODIS dataset simply captures the boundary layer state at around 1:30pm. This will need to be considered in

future studies using the AMSR-MODIS dataset as with any other dataset measuring any atmospheric parameter with diurnal cycle.

The AMSR and MODIS equator crossing time will be mention in the following sentence: All these instruments orbit in tandem measuring the same volume of air within minutes of each other, that is, by design, these measurements are collocated; **their equatorial crossing time is ~1:30pm.**

p.8, l.23-25: This sentence (That is version2.0 (…) algorithm) is not necessary in the summary.  The sentence will be deleted from the summary.

Figure 8: Please provide information on the monthly variation of BL-CWV and BL-CTH, e.g. showing error bars or box-and-whisker plots

Below is the figure showing the standard deviation.

[Figure]

To avoid cluttering, the following figure will be used with the following sentence in the caption: The numbers shown are the average standard deviation per region.

[Figure]

Figure 9: Since you plot normalized values, the unit [mm] is not correct!  Thanks for spotting this, it will be changed to mm mm-1.

**Technical corrections:**

p.1, l.20: replace "are" by "is"  Done
p.1, l.24: "processes"    Done
p.3, l.22: "criterion", not criteria  We believe that criteria is the right option since we use a temporal condition, a spatial condition, and, in the case of GPS-RO, a sharpness parameter condition.
p.3, l.24: "gridding"  Done
p.3, l.25: "represent" (not "represents")  Done
p.5, l.1: "coast" (not "cost")  Done
p.6, l.31, p.7, l.10: "4 K" _(without degree sign)  Done
p.8, l.14: "remain", not "remains"  Done
p.8, l.32: "over", not "on"  Done
p.8, l.32: "Sc-Cu": You never introduced these acronyms:  we changed to Stratocumulus to Cumulus
Fig. 9 (caption): please correct: Normalize**d** (…) The number**s** (…) coefficient**s** (…) Done

**References:**

Horvath et al (2013) - doi:10.1002/2013JD021355
Grosvenor et al (2014) - doi:10.5194/acp-14-7291-2014
Marchant et al. (2016) – doi:10.5194/amt-9-1587-2016